# Synthesis and Characterization of Na₃SbS₄ Solid Electrolytes via Mechanochemical and Sintered Solid-State Reactions: A Comparative Study

**Celastin Bebina Thairiyarayar [1,†], Chia-Hung Huang [2,3,†], Yasser Ashraf Gandomi [4] , Chien-Te Hsieh [5,6,*] and Wei-Ren Liu [1,*]**

1   Department of Chemical Engineering, R&D Center for Membrane Technology,
    Chung Yuan Christian University, 200 Chung Pei Road, Chungli District, Taoyuan City 32023, Taiwan;
    g11002101@cycu.edu.tw
2   Department of Electrical Engineering, National University of Tainan, No. 33, Sec. 2, Shulin St., West Central
    District, Tainan City 700, Taiwan; chiahung@mail.mirdc.org.tw
3   Metal Industries Research and Development Centre, Kaohsiung 701, Taiwan
4   Department of Chemical Engineering, Massachusetts Institute of Technology, Cambridge, MA 02142, USA;
    ygandomi@mit.edu
5   Department of Chemical Engineering and Materials Science, Yuan Ze University, Taoyuan City 32003, Taiwan
6   Department of Mechanical, Aerospace, and Biomedical Engineering, University of Tennessee,
    Knoxville, TN 37996, USA
*   Correspondence: cthsieh@saturn.yzu.edu.tw (C.-T.H.); wrliu@cycu.edu.tw (W.-R.L.);
    Tel.: +886-3-265-4140 (W.-R.L.); Fax: +886-3-265-4199 (W.-R.L.)
†   These authors contributed equally to this work.

**Abstract:** A sulfide-based solid electrolyte is an enticing non-organic solid-state electrolyte developed under ambient conditions. Na₃SbS₄, a profoundly enduring substance capable of withstanding exceedingly elevated temperatures and pressures, emerges as a focal point. Within this investigation, we employ dual distinct techniques to fabricate Na₃SbS₄, encompassing ball milling and the combination of ball milling with sintering procedures. A remarkable ionic conductivity of $3.1 \times 10^{-4}$ S/cm at room temperature (RT), coupled with a meager activation energy of 0.21 eV, is achieved through a bifurcated process, which is attributed to the presence of tetragonal Na₃SbS₄ (t-NSS). Furthermore, we delve into the electrochemical performance and cyclic longevity of the Na₂/₃Fe₁/₂Mn₁/₂O₂ | t-NSS | Na system within ambient environs. It reveals 160 mAh/g initial charge and 106 mAh/g discharge capacities at 0.01 A/g current density. Furthermore, a cycle life test conducted at 0.01 A/g over 30 cycles demonstrates stable and reliable performance. The capacity retention further highlights its enduring energy storage capabilities. This study underscores the sustainable potential of Na₃SbS₄ as a solid-state electrolyte for advanced energy storage systems.

**Keywords:** t-Na₃SbS₄; mechanochemical (BM); sintering; ionic conductivity; activation energy

## 1. Introduction

Solid electrolytes (SEs) are better separators for battery safety problems than liquid electrolytes [1,2]. Sulfide and selenide [3–5], oxide [6,7], β alumina [8,9], and NASICON [10,11], borohydride [12,13], and halo-aluminates [14] are the types of inorganic solid electrolytes which have been used so far as the solid electrolytes in sodium-ion batteries. Due to high ionic conductivity, low cost, good mechanical properties, wide electrochemical window, and formation of stable solid electrolyte interphase (SEI), sulfides received extensive research attention as one of the inorganic solid electrolytes in comparison to the other solid electrolytes [5,15]. Hayashi introduced Na₃PS₄ as one of the most well-known sulfide SE in all-solid-state sodium-ion batteries by substituting sodium for lithium in Li₂S-P₂S₅ [3,16,17]. This material exhibits high ionic conductivity at room temperature (RT). It has also been

investigated for the replacement of phosphorus by antimony as a $Na_3SbS_4$ [18,19]. The $Na_3SbS_4$ material was synthesized using various approaches, such as solid-state methods, wet processes [5,20,21], mechanochemical processes and then sintering [22–24], sintering and then mechanochemical processes [18,25,26], and sintering alone [27–30]. Different synthesis techniques affected its ionic conductivities and activation energies. Our primary research focus is on the synthesis of $Na_3SbS_4$, a sulfide solid electrolyte customized for application in sodium-ion batteries. This choice is motivated by the manifold advantages that sulfide electrolytes offer over their liquid counterparts, including enhanced battery safety and superior ionic conductivity [5,15]. An extensive literature review was conducted to comprehensively address critical aspects, such as ionic conductivity and electrochemical performance. Through our rigorous examination of various synthesis methods, it became evident that certain approaches proved remarkably effective.

For instance, Banerjee et al. reported the synthesis of $Na_3SbS_4$ powder using two different methods. The first method involved a solid-state process, where a mixture of $Na_2S$, $Sb_2S_3$, and elemental sulfur was heat-treated at different temperatures in a vacuum-sealed quartz ampoule. The second method involved a solution-based process, where the solid powders were dissolved in anhydrous methanol or deionized water. The resulting solution was dried and heat-treated at different temperatures under a vacuum. The resulting $Na_3SbS_4$ solid electrolyte exhibited a high conductivity of 1.1 mS cm$^{-1}$ at 25 °C and a low activation energy of 0.20 eV [5].

Rush et al. synthesized tetragonal $Na_3SbS_4$ using a ball milling and sintering method. The synthesis involved ball milling 4 g of $Na_3SbS_4 \cdot 9H_2O$ with $ZrO_2$ balls for 5 min, then heating the powder under vacuum at a ramp rate of 5 °C/min to 150 °C for 1 h. The resulting powder was transferred to an Argon-filled glove box for further processing. The synthesized tetragonal $Na_3SbS_4$ exhibited an impressive ionic conductivity of over 0.8 mS cm$^{-1}$ at room temperature (RT). In comparison, Na-ion migration in tetragonal $Na_3SbS_4$ has an activation energy of 0.3 eV, while Na-ion migration in cubic $Na_3SbS_3$ has an activation energy of 0.5 eV [22]. Wu et al. successfully synthesized $Na_3SbS_4$ (NSS) through sintering and ball milling. They mixed $Na_2S$, Sb powder, and S in a ratio and pressed the mixture into a pellet. The pellet was heat-treated in a box furnace and ball-milled for 20 h. The resulting NSS material had a tetragonal crystal structure with an ionic conductivity of 1.06 mS cm$^{-1}$ at room temperature (RT) and an activation energy of 0.216 eV. These results suggested that NSS has high ionic mobility, making it a promising candidate for use in solid-state batteries. This study highlights the effectiveness of solid-state synthesis and ball milling for producing high-performance solid electrolytes [26]. Wang et al. investigated the relationship between pressure loading and the ionic conductivity of $Na_3SbS_4$ solid electrolytes prepared by sintering alone. The starting material, $Na_3SbS_4 \cdot 9H_2O$, was sintered at 150 °C under a vacuum and held for 1 h. The researchers found that the ionic conductivity of $Na_3SbS_4$ increased as pressure loading increased, mainly due to a decrease in grain boundary resistance. As the pressure gradually increased from 0.4 GPa to 0.9 GPa, the ionic conductivity increased from 0.7 mS cm$^{-1}$ to 1 mS cm$^{-1}$ and then tended to be constant beyond. Interestingly, the conductivity further increased during decompression due to decreased bulk and grain boundary resistance. After the pressure was released to 0.1 GPa, the impedance spectra were collected, and the ionic conductivity was found to be stable at around 1.6 mS cm$^{-1}$ for 24 h. This work demonstrated that pressure can effectively increase the ionic conductivity of sulfide-based solid electrolytes [30]. Typically, ball milling is a type of grinding where chemicals are combined to create a fine powder while reducing particle size and enhancing material reactivity. There are no adverse reactions, and it has a wide range of applications and is suitable for continuous usage [31,32]. The mechanical characteristics were enhanced, and its porosity was reduced throughout the sintering process [33,34].

As far as the current literature is concerned, limited research specifically explores using ball milling and sintering methods to synthesize $Na_3SbS_4$ solid electrolytes with high ionic conductivity and low activation energy for sodium-ion batteries. Thus, in this study,

we attempt to synthesize the t-$Na_3SbS_4$ using two methods, the first being ball milling at 5 to 20 h, the second being ball milling at 20 h, and then sintering at three different temperatures at 12 h. Further, we compare their results with characterizations of materials of the crystal phase structure and its properties from X-ray diffraction techniques (XRD) and Rietveld refinement. The surface morphology of the materials and electronic state of the atoms is obtained using scanning electronic microscopy/energy dispersive X-ray spectroscopy (SEM/EDX), X-ray Photoelectron Spectroscopy (XPS), and electrochemical performance of ionic conductivity. The activation energy is characterized with electron impendence spectroscopy (EIS). The evaluation of the performance and sensitivity of a sensor in detecting hydrogen sulfide gas is carried out by an $H_2S$ sensor test to demonstrate air stability of as-synthesized NSS solid electrolyte. The corresponding electrochemical tests, such as Galvanostatic charge/discharge tests and rate capability tests, are carried out by using $Na_{2/3}(Fe_{1/2}Mn_{1/2})O_2$/NSS/Na coin cells. This study not only aims for technical advancements but also emphasizes the sustainability of the synthesized solid electrolyte.

## 2. Materials and Methods

### 2.1. Materials and Chemicals

Sodium sulfide ($Na_2S$), a powder with a purity of 90%, was purchased from Alfa Aesar in Karlsruhe, Germany. The Antimony (III) sulfide ($Sb_2S_3$), with a purity of 98%, was also purchased from Alfa Aesar in Germany. Additionally, we obtained sulfur with a purity of 99.9% from Sigma-Aldrich in Schnelldorf, Germany.

### 2.2. Synthesis of $Na_3SbS_4$ Solid Electrolyte

The two methods for synthesizing $Na_3SbS_4$ (NSS) are shown in Scheme 1: ball milling (BM) alone and ball milling plus sintering (BM-H). $Na_2S$, $Sb_2S_3$, and sulfur were the starting components in the first process, in a 3:1:2 ratio. $ZrO_2$ balls with a thickness of 10 balls/10 mm were used to ball mill the mixture for 20 h at 510 rpm. $Na_3SbS_4$, also known as NSS-BM-20h, was the end product. In the second process, the final product from the first approach served as the starting material. It was sintered for 12 h under vacuum at three different temperatures (250, 300, and 350 °C) with a heating rate of 5 °C/min, resulting in samples designated as NSS-BM-20h-H250, NSS-BM-20h-H300, and NSS-BM-20h-H350 (also known as NSS-BM-H250, NSS-BM-H300, and NSS-BM-H350, respectively). To preserve the samples' purity, the whole synthesis and post-processing stages were completed in a glovebox (LABstar, MBraun, München, Germany, $H_2O$ and $O_2$ < 0.5 ppm). Additionally, we extended the ball milling time to 25 h and synthesized samples at lower temperatures of 150 and 200 °C, which are labeled as NSS-BM-25 h, NSS-BM20h-H150, and NSS-BM20h-H200, respectively.

### 2.3. Material Characterization

The crystallographic phase was identified using X-ray powder diffraction (XRD) on a Bruker eco D8 advance diffractometer with monochromatic CuK radiation (λ = 1.54060 Å). The samples were analyzed using closed XRD plates, and XRD data were collected at a scan rate of 2°min$^{-1}$ in steps of 0.02° in the range of 10–80°. Rietveld structure refinement against XRD data was performed using GSAS–II (1.0.0.) software. The crystal structure was determined using the Diamond-Crystal and Molecular Structure Visualization software Version 3.2. Quantitative elemental analyses and elemental distributions were obtained using energy dispersive spectrometry and electron mapping, while images of the morphology and microstructure of the materials were captured using field emission scanning electron microscopy (FE-SEM, JSM-7600F; JEOL) (EDS, X-MAX, Oxford Instrument, Abingdon, UK). The K-Alpha XPS spectrometer (Thermo Scientific, Waltham, MA, USA) was used to conduct X-ray photoelectron spectroscopy (XPS) to determine the chemical valence states. A specific (Riken Keiki HS-04) $H_2S$ sensor was used to assess the air stability of NSS. A 180 mg NSS electrolyte pellet was prepared and pressed at a pressure of 360 MPa. The pellet was carefully inserted into a 12 cm × 12 cm × 12 cm airtight acrylic box from the glovebox.

Consequently, the $H_2S$ sensor was placed inside the container, and measurements were made every 30 s while the sensor readings were being recorded. This experimental design made it possible to monitor $H_2S$ gas concentrations and offered insightful information on the sensor's receptivity and sensitivity to the target gas.

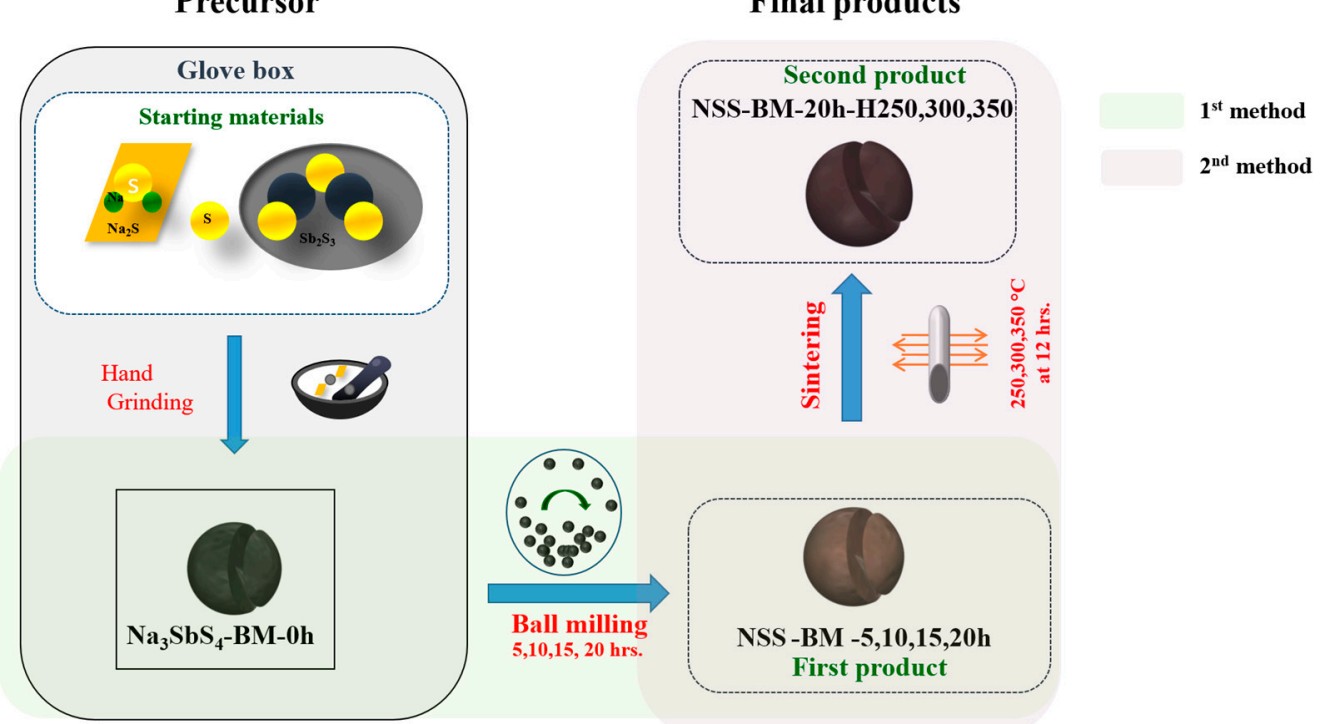

**Scheme 1.** Synthesis of $Na_3SbS_4$ powder with mechanochemical (BM) and BM processes followed by sintering process.

### 2.4. Electrochemical Characterization

Using an E.C. Lab, Biologic Potentiostat, model SP-200, experiments were performed to measure the samples' ionic conductivity and activation energy in the frequency range of 7 MHz–10 Hz at an A.C. amplitude of 10 mV using electrochemical impedance spectroscopy (EIS). With the help of the KP cell, ionic conductivities and electrochemical behaviors were explored, respectively. A PTFE tank with a diameter of 10 mm was used to assemble the KP cell. About 70 mg of NSS powder was placed inside, and it was cold-pressed into a pellet for one minute at a pressure of 360 MPa to test the ionic conductivity and activation energy of the NSS samples.

### 2.5. Cell Assembly for Sodium-Ion Battery Testing

In sodium-ion battery testing, a half-cell configuration was employed to assess the electrochemical properties. The electrode slurry, consisting of NFMO ($Na_{2/3}Fe_{1/2}Mn_{1/2}O_2$), PVDF and Super P in an 8:1:1 ratio, was thoroughly mixed overnight for homogeneity. It was then coated onto an aluminum foil substrate and subjected to vacuum heating at 120 degrees Celsius for 9 h to fabricate the electrode. The resulting electrode was transferred to a glovebox for further assembly. The cell assembly involved pressing a 70 mg of NSS electrolyte into a pellet under a pressure of 360 MPa, which was then sandwiched between two electrodes, completing the half-cell setup.

Additionally, a drop of 1M $NaClO_4$ solution into a propylene carbonate (PC, Sigma-Aldrich, St. Louis, MO, USA) and 5% fluoroethylene carbonate (FEC, Sigma-Aldrich) was applied to enhance the contact between the cathode and solid electrolyte. Na metal was used as the anode material. The cell assembly was performed in an argon-filled glovebox with controlled oxygen and water levels. Galvanostatic charge–discharge (GCD)

analysis was conducted at different current densities of 0.01, 0.02, and 0.05 A/g within a potential window of 2.0 V to 3.8 V vs. Na/Na$^+$. The GCD analysis was carried out using a Neware multichannel battery tester at room temperature (RT) to evaluate the Na-ion storage capacity and cyclic stability. This comprehensive approach allowed for the systematic evaluation of the electrochemical performance of the sodium-ion battery system under controlled conditions.

## 3. Results and Discussion

Scheme 1 shows the flow chart of how we synthesized NSS by using two different methods. The precursors were ball-milled for 5, 10, 15, and 20 h, with the rationale being to investigate the effect of milling time on the resulting particle size and crystallite size. Figure 1a shows the XRD patterns of NSS at different ball milling time. The peak noted at 17° exhibited substantial intensities, while the presence of the tetragonal phase augmented in correlation with the duration of milling. Following 20 h of ball milling, the emergence of a tetragonal phase was discerned and subsequently verified through the standard pattern of ICSD 47291 and another pertinent prior reference [27]. The insets in Figure 1a display photo images of NSS-BM-0 h, NSS-BM-5 h, NSS-BM-10 h, NSS-BM-15 h, and NSS-BM-20 h, respectively. In Figure 1b, particular X-ray diffraction (XRD) analysis was shown in an expanded form of 15° to 20° to observe the peaks between 17° to 18°. One of the tetragonal peaks emerged at 17.5° after 10 h of ball milling, evolving from the initial peaks at 15.9° (Na$_2$Sb) and 17.8° (sulfur) presented in the precursor. This indicated structural modifications and potential phase transformations induced by the extended milling process. Further increasing the ball milling time to 20 h resulted in splitting the single peak into two narrow, smaller peaks. These observations provided insights into the structural changes and crystallinity of the material as a result of different ball milling durations. Figure 1c displays the Rietveld refinement analysis of NSS after 20 h. The result showed that, after 20 h ball milling, two phases were presented, with the main phase being tetragonal Na$_3$SbS$_4$ (P-421c) at 80.6%, with 26.9, 31.2, and 44.7 degrees 2θ peak from cubic NaSbS$_2$ (Fm-3m) at 19.4%. The R$_p$ and R$_{wp}$ values obtained from the refinement were 14.16% and 18.73%, respectively. In Figure 1d, the grain size of Na$_3$SbS$_4$-0 h that was subjected to 20 h of ball milling was calculated using the Scherer equation [35]. The grain sizes of NSS-BM-0 h, NSS-BM-5 h, NSS-BM-10 h, and NSS-BM-20 h were determined to be 220, 218, 203, and 196 nm, respectively. The results indicated that the crystallite size of NSS decreased from 220 to 196 nm when increasing the milling time from 0 h to 20 h. Supplementary Figure S2a displays the X-ray diffraction (XRD) patterns and corresponding photographs of samples subjected to 20 and 25 h of ball milling. An interesting observation is the emergence of distinct 2θ peaks around 16.6° in BM-25 h, indicating the presence of NaSbS$_2$. This suggests a transformation or phase transition within the material due to the extended ball milling process. This alteration may be attributed to enhanced structural rearrangements and increased defects induced by prolonged milling, leading to the formation of NaSbS$_2$.

Figure 2a–d display SEM images of NSS-BM-0 h, NSS-BM-10 h, NSS-BM-15 h, and NSS-BM-20 h, respectively. In Figure 2, the SEM image shows a comparison between the precursor (NSS-BM-0 h) and the specific particle sizes obtained in the range of 1 μm at each milling interval. After undergoing ball milling for a duration of 10 h, the resultant particle size was measured to be within the range of 49–26 μm. Similarly, upon extending the milling time to 15 h, the particle size exhibited a distribution of 10–20 μm, and further increasing the milling time to 20 h led to a particle size range of 1–10 μm. The unmistakable inference drawn from these results is that the ball milling technique significantly and distinctly diminished the grain sizes of the particles, albeit in an irregular and unpredictable manner.

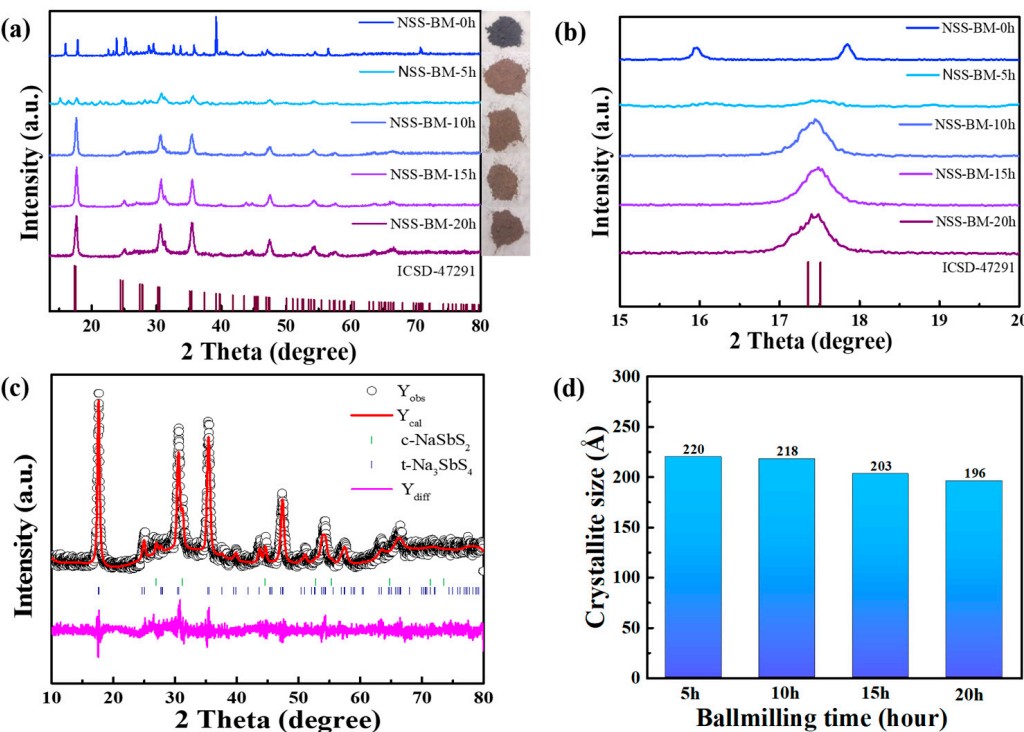

**Figure 1.** (**a**) X-ray diffraction patterns of BM-only $Na_3SbS_4$ with respective sample photos. (**b**) Magnified version of main peak from 15 to 20° at 2 θ. (**c**) Rietveld refinement profile of NSS-BM-20 h using Bragg positions as t-$Na_3SbS_4$ and c-$NaSbS_2$. (**d**)The crystallite size of ball milling sample as an hours function.

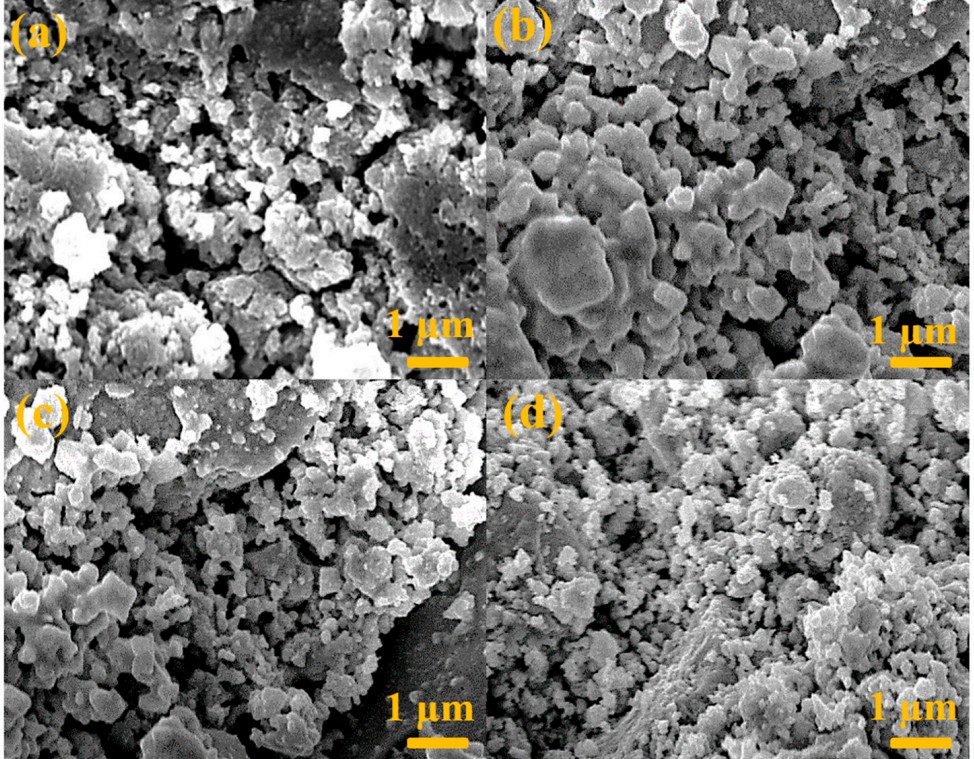

**Figure 2.** SEM images of (**a**) NSS precursor (NSS-BM-0 h), (**b**) NSS-BM-10 h, (**c**) NSS-BM-15 h, and (**d**) NSS-BM-20 h.

In Figure 3a, XRD patterns of NSS-BM-H250, NSS-BM-H300, and NSS-BM-H350 are shown alongside the standard Na$_3$SbS$_4$ pattern (#47291, ICSD). Insets in Figure 3a display sample images. Samples exhibited a higher intensity and a tetragonal phase with rising sintering temperature, and were darkened in color. The XRD analysis confirms tetragonal peaks for NSS-BM sintered at 250 °C, 300 °C, and 350 °C, which align with the standard pattern. Figure 3b charts show the NSS-BM grain size evolution via the Scherer equation as a function of sintering temperature. Crystallite size increases (250 °C: 273 nm, 300 °C: 297 nm) and then decreases (350 °C: 208 nm). NSS-BM-250, milled for 20 h and sintered at 250 °C, is the largest in size (273 nm) compared to NSS-20 h (196 nm). Supplementary Figure S3a showcases the X-ray diffraction (XRD) patterns along with accompanying photographs of NSS-BM20h-H150, 200, and 250 samples. Notably, the XRD pattern of NSS-BM20h-H150 exhibits several minor impurity peaks. This occurrence can be attributed to the lower sintering temperature (150 °C), which may not provide sufficient energy to promote complete phase transformation and crystallization. Consequently, residual impurities persist in the sample, leading to the observed minor peaks in the XRD pattern.

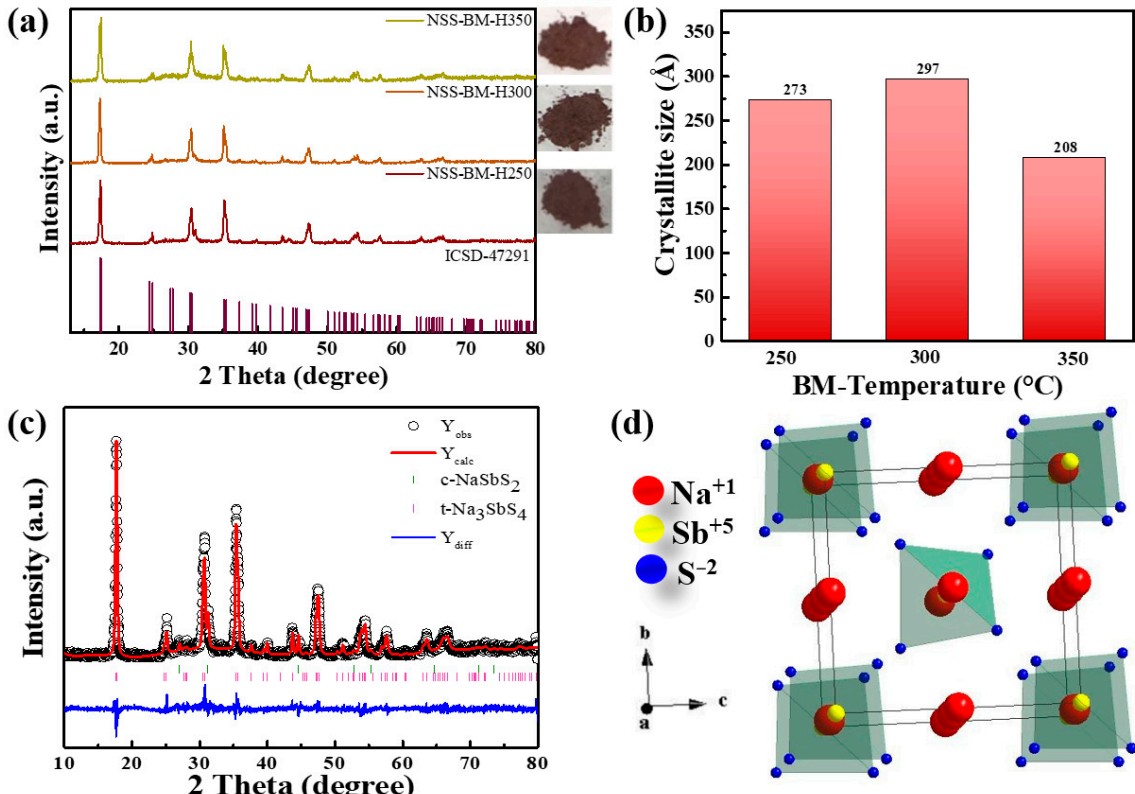

**Figure 3.** (**a**) X-ray diffraction patterns of BM process and then sintering Na$_3$SbS$_4$ (BM-H), with respective sample photos. (**b**) The crystallite size of NSS BM-H sample as a function of different temperatures. (**c**) Rietveld refinement profile of NSS-BM20h-H250 using Bragg positions as t-Na$_3$SbS$_4$ and c-NaSbS$_2$. (**d**) Crystal structure of t-Na$_3$SbS$_4$-BM20h-H250.

Rietveld refinement in Figure 2c shows a cubic phase decrease with a higher sintering temperature. The 20 h ball-milled sample sintered at 250 °C has the highest cubic phase content. This intricate interplay between ball milling, sintering, and crystallographic phases influences material properties. Figure 3d shows the crystal structure of Na$_3$SbS$_4$-BM-H250, which possesses a tetragonal crystal system and is a member of the space group P-42$_1$c, with R$_p$ at 14.16% and R$_{wp}$ at 18.73%. The lattice constants were determined to be a = 7.18392(0) Å and c = 7.26153(2) Å.

The Na1 site in 4d, the Na2 site in 2b, the Sb1 site in 2a, and the S site in 8e are identified as the four unique sites in the crystal structure. Na ions are surrounded by the $SbS_4{}^{3-}$ tetrahedral unit. The crystallographic details are listed in Table 1.

**Table 1.** Crystal structure information of $Na_3SbS_4$-BM20h-H250. Crystal system—tetragonal lattice parameters: A = 7.1839(20) Å. C = 7.2615(32) Å. Space group: P-42$_1$c (114). Volume, Z: V = 373.08(6) Å$^3$, Z = 2.

| Atoms | Wyckoff | Occupancy | x/a | y/a | z/c |
|---|---|---|---|---|---|
| Na1 | 4d | 1 | 0 | 1/2 | 1/2 |
| Na2 | 2b | 1 | 0 | 0 | 1/2 |
| Sb | 2a | 1 | 0 | 0 | 0 |
| S | 8e | 1 | 0.29331 | 0.33080 | 0.68745 |

$R_{wp}$ = 18.73%; $R_p$ = 14.16%; $\chi^2$ = 1.29.

Figure 4a–d display the SEM images of NSS-BM-20 h, NSS-BM-H250, NSS-BM-H300, and NSS-BM-H350, respectively. The surface morphology of NSS-BM-20 h, shown in Figure 4a, revealed an irregular form in the range of 1 μm. Subsequently, sintering the sample at different temperatures induced substantial changes in its morphology. With increasing sintering temperatures, there was a notable increase in the neck formation between particles, resulting in the particle merging at NSS-BM-H250 (10–16 μm), 300 °C (13–45 μm), and obtaining a dense morphology at 350 °C. Moreover, the impact of the sintering temperature on the surface morphology and crystal structure of $Na_3SbS_4$ (NSS) is noteworthy. This influence is evidenced by the substantial crystal size augmentation observed in NSS-BM-H250, which can be attributed to the distinct grain sizes in NSS-BM-20 h (ranging from 1 to 10 μm) and NSS-BM-H250 (ranging from 10 to 16 μm). The sintering temperature acts as a critical factor in determining these variations. As the sintering temperature increases, the particles experience enhanced diffusion, leading to stronger bonding between adjacent particles. This phenomenon promotes particle coalescence and densification of the material.

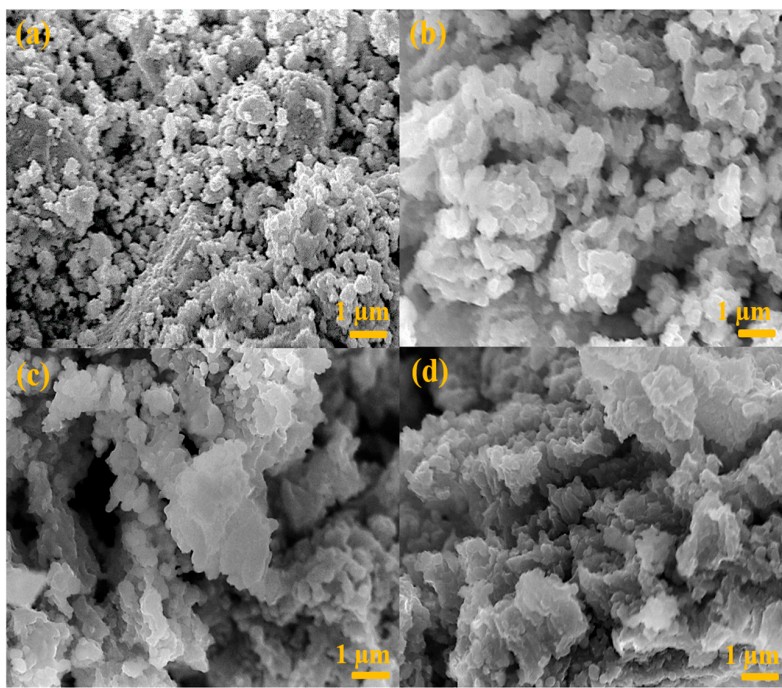

**Figure 4.** SEM images of (**a**) NSS-BM-20 h as precursor, (**b**) NSS-BM-H250, (**c**) NSS-BM-H300, and (**d**) NSS-BM-H350.

The SEM/EDX mapping of NSS-BM-20 h (a)–(d) and NSS-BM-20h–H250 (e)–(h) is displayed in Figure 5. The elements Na, Sb, and S were evenly distributed across the sample, according to the four sections of the mapping. The SEM/EDX mapping, which was thoroughly approached, explains the synthesis of NSS samples. Processing was likely carried out under ideal conditions, resulting in a highly pure and homogenous product, as seen by the elements' remarkably uniform appearance.

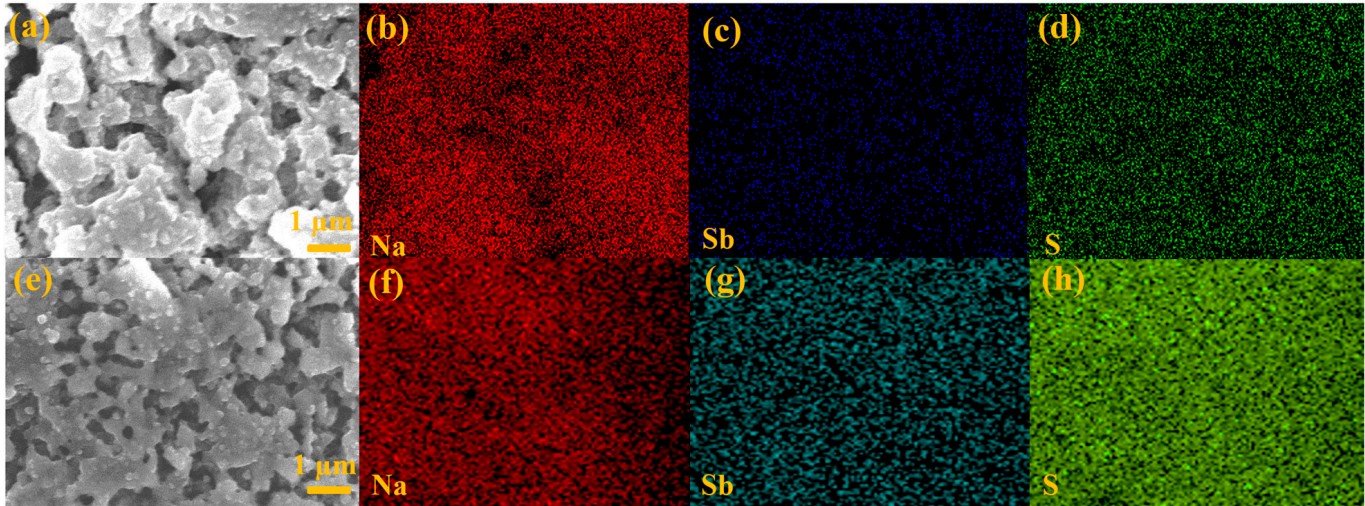

**Figure 5.** SEM/EDX mapping of (**a**–**d**) NSS-BM-20 h and (**e**–**h**) NSS-BM-20h–H250.

Figure 6a–h show the XPS spectra of the NSS materials with the BM (NSS-BM-20 h) and BM processes, which were then sintered at different temperatures (NSS-BM-20h-H250, NSS-BM-20h-H300, NSS-BM-20h-H350). The XPS results revealed two single signals for $SbS_4{}^{3-}$, with no shift observed in the Sb 3d$^3$ spectra of NSS-BM-20 h (Figure 6a), NSS-BM-20h-H250 (Figure 6c), NSS-BM-20h-H300 (Figure 6e), and NSS-BM-20h-H350 (Figure 6g). The increase in the O 1s signal with decreasing temperature was due to the removal of the adsorbed species and an increase in the stoichiometric ratio of Na/Sb in the sample. The S 2p spectra of NSS-BM-20 h in Figure 6b and NSS-BM-20h-H250 in Figure 6d, NSS-BM-20h-H300 in Figure 6f, and NSS-BM-20h-H350 in Figure 6h showed a Sb-Na-S signal, which was identified as a single doublet signal. The S 2p spectra, on the other hand, revealed no impurity signals, indicating that the samples were highly pure. The slightly lower signal observed in NSS-BM-20h-H250, compared to NSS-BM-20 h, could be due to the formation of surface defects or changes in the oxidation state of the sulfur species, which was confirmed by a previous study [23].

Figure 7a shows the ionic conductivity of NSS BM-20 h, NSS BM-20h-H250, 300, and 350 °C at room temperature (RT). The corresponding resistance values are provided in Table 2. Figure 7a and Table 2 show the BM-20 h total grain boundary resistance increasing from 176 Ω to 202 Ω in BM-20h-H250, indicating a decrease of bulk resistance ($R_{bulk}$), grain boundary resistance ($R_{gb}$), and total resistance ($R_{total}$) of the samples, and an increase in sintering temperature, resulting in a decrease in ionic conductivity. The inset in Figure 7a displays the corresponding equivalent circuit. However, the BM-20h-H250 sample showed the highest ionic conductivity ($3.1 \times 10^{-4}$ S cm$^{-1}$). Interestingly, it is identical to a previous study, in which they used direct $Na_3SbS_4 \cdot 9H_2O$ as a starting material [27].

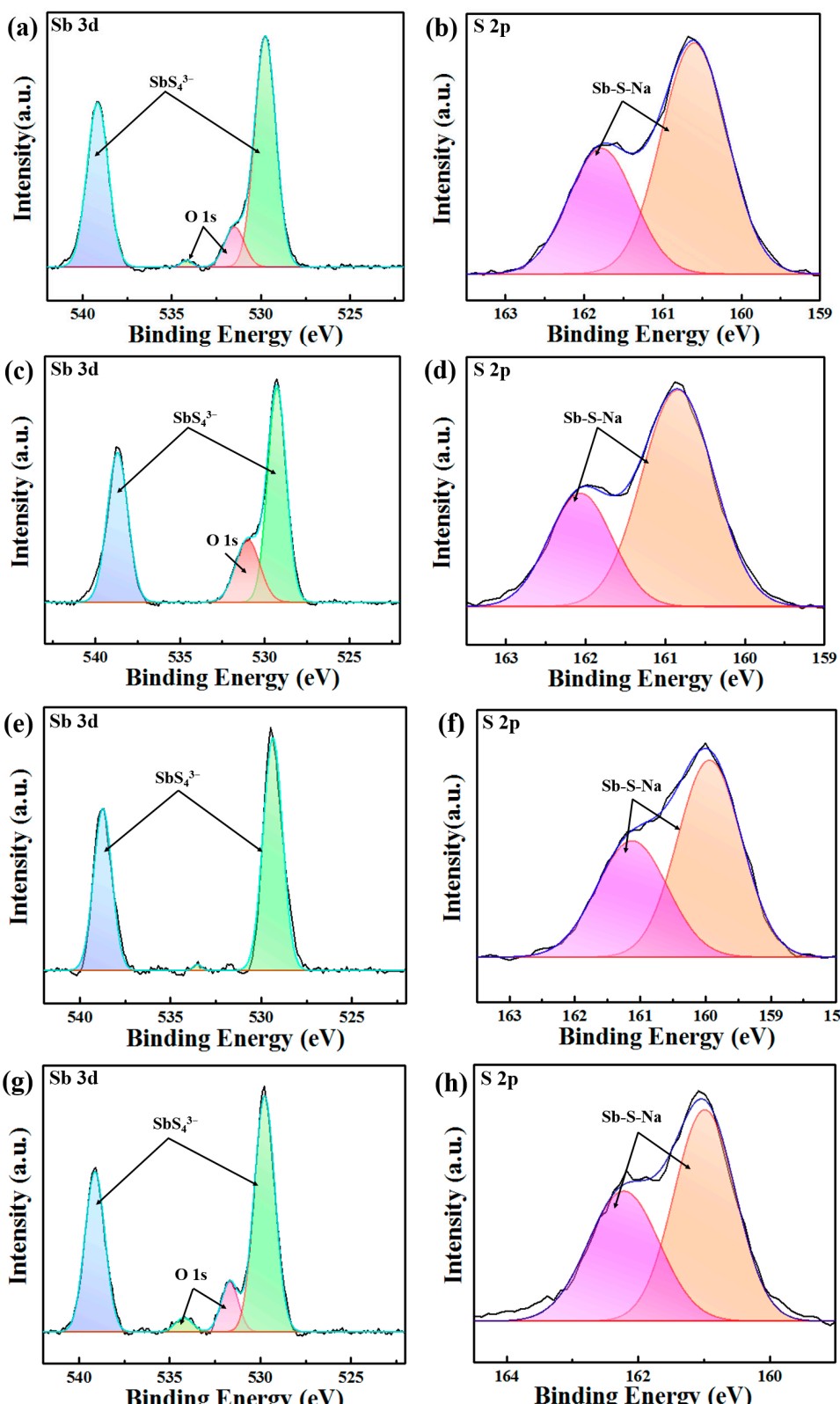

**Figure 6.** High-resolution XPS spectra of NSS with BM process, and BM process and then different temperatures: (**a**) Sb 3d3 spectra and (**b**) S 2p spectra of NSS-BM-20 h; (**c**) Sb 3d3 spectra and (**d**) S 2p spectra of NSS-BM-20h-H250; (**e**) Sb 3d3 spectra and (**f**) S 2p spectra of NSS-BM-20h-H300; (**g**) Sb 3d3 spectra and (**h**) S 2p spectra of NSS-BM-20h-H350.

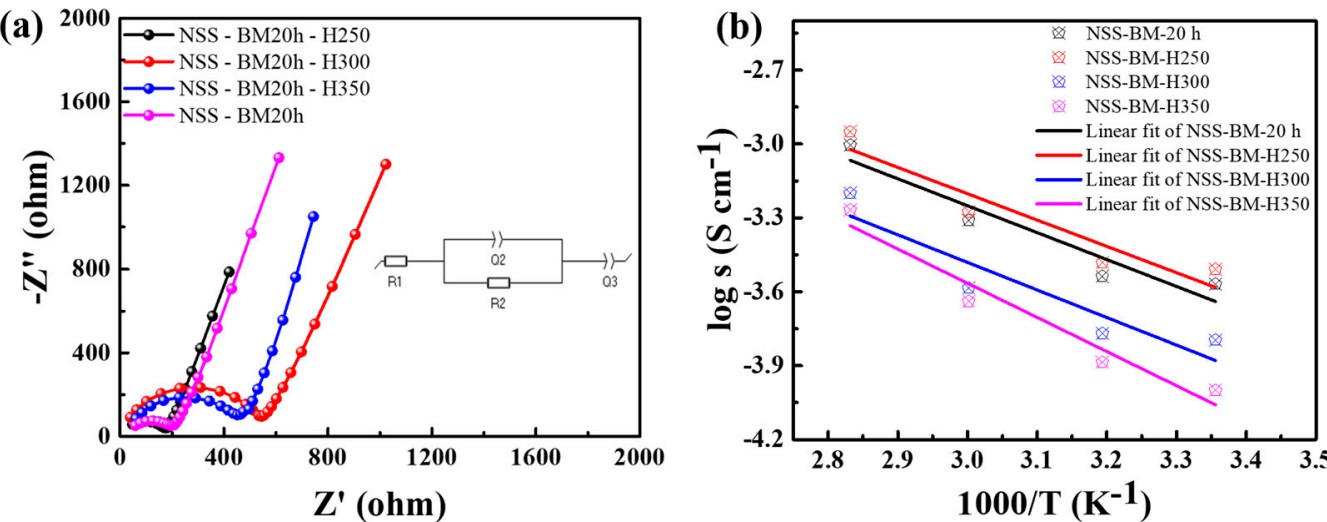

**Figure 7.** (**a**) Nyquist impedance plots at RT: equivalent circuit in the system and (**b**) Arrhenius plots at 25, 40, 60, and 80 °C of NSS-BM20 h and NSS BM-H250, 300, and 350 °C.

**Table 2.** Comparison of ionic conductivity and activation energy of $Na_3SbS_4$ with BM process, and BM process and then sintering at different temperatures.

| Samples | L (cm) | S (cm$^2$) | $R_b$ (Ω) | $R_{gb}$ (Ω) | $R_b + R_{gb}$ (Ω) | $\sigma_{25\,°C}$ (S cm$^{-1}$) | $E_a$ (eV) |
|---------|--------|------------|-----------|--------------|--------------------|---------------------------------|------------|
| NSS-BM-20 h | 0.044 | 0.7854 | 31 | 171.4 | 202.4 | $2.77 \times 10^{-4}$ | 0.22 |
| NSS-BM-H250 | 0.043 | 0.7854 | 11.43 | 164.7 | 176.13 | $3.11 \times 10^{-4}$ | 0.21 |
| NSS-BM-H300 | 0.043 | 0.7854 | 13.38 | 518.9 | 532.28 | $1.03 \times 10^{-4}$ | 0.23 |
| NSS-BM-H350 | 0.039 | 0.7854 | 26.55 | 439.2 | 465.75 | $1.07 \times 10^{-4}$ | 0.28 |

Figure 7b shows the Arrhenius plots of NSS BM-20h, NSS BM-20h-H250, 300, and 350 samples at 25, 40, 60, and 80 °C, respectively. The BM-20h-H250 sample exhibited the lowest activation energy (0.21), and its ionic conductivity was 0.31 m S cm$^{-1}$ at 25 °C and 1.12 m S cm$^{-1}$ at 80 °C. The results further confirmed that a low grain boundary leads to high ionic conductivity and low activation energy. The equivalent circuit of the system and the Arrhenius equation elucidated the activation energy, which was estimated using Equation (1), as follows:

$$\sigma = A \exp(-E_a/kT) \tag{1}$$

where A is the pre-exponential factor, T is the absolute temperature, and *k* is the Boltzmann constant, i.e., the absolute temperature [36]. The comprehensive dataset presented in Table 2 furnishes substantiation for the derived conclusions and delineates the influential facets governing the ionic conductivity and activation energy of the NSS BM-20 h and NSS BM-20h-H250, 300, and 350 samples. These factors include bulk resistance ($R_{bulk}$), grain boundary resistance ($R_{gb}$), and total resistance ($R_{total}$).

Supplementary Figure S1a offers a clear depiction of the ionic conductivity plot, showcasing the effects of varying ball milling durations (10 h, 15 h) on NSS samples. As the duration of ball milling increases, there is a marked reduction in total resistance. This reduction reaches its zenith at the 20 h mark, resulting in a substantial boost in ionic conductivity. These enhancements can be attributed to the refinement in the crystal structure and the diminishment of particle size. Together, these elements synergize to amplify ion mobility and effectively suppress resistance. Supplementary Figure S1b complements this observation by presenting the activation energy corresponding to ball milling durations of 10 h, 15 h, and 20 h. It is worth noting that an extension in ball milling time leads to a reduction in activation energy, signifying an improvement in ionic conductivity. For a detailed breakdown of the ionic conductivity and activation energy values, please refer to

Supplementary Table S1. In contrast, at the 25 h mark (Supplementary Figure S2), there is a notable surge in the total resistance from 202 $\Omega$ to 403 $\Omega$, resulting in a significant drop in ionic conductivity from 2.77 to 1.32 $\times$ 10$^{-4}$ S cm$^{-1}$. This decline can be ascribed to the prolonged milling process, which encourages particle agglomeration and carries the potential for crystal damage. Consequently, this leads to an overall increase in total resistance and a corresponding decrease in ionic conductivity.

Furthermore, Supplementary Figure S3b imparts valuable insights into the impact of the sintering temperature on ionic conductivity, with a specific focus on NSS BM-20 h samples sintered at 150 °C, 200 °C, and 250 °C. A conspicuous trend emerges: as sintering temperature diminishes, so does ionic conductivity. This phenomenon finds its roots in the diminished thermal energy at lower temperatures, which effectively curtails ion mobility, consequently resulting in a discernible reduction in ionic conductivity. Detailed ionic conductivity values are provided in Supplementary Table S2.

Figure 8 portrays the air stability of the $Na_3SbS_4$-BM-H250 solid electrolyte. Remarkably, the results divulged the absence of any conspicuous reaction discernible within the $H_2S$ sensor during an extended period that exceeded 45 min, even in the presence of elevated humidity levels (56%) after air exposure at room temperature (RT). This captivating disclosure underscores the enigmatic reality that the sulfides within the $Na_3SbS_4$ solid electrolyte remain impervious to atmospheric circumstances, effectively impeding the genesis of $H_2S$ gas. This sustainable characteristic showcases the material's praiseworthy endurance and bolsters its safety paradigm [19,37,38].

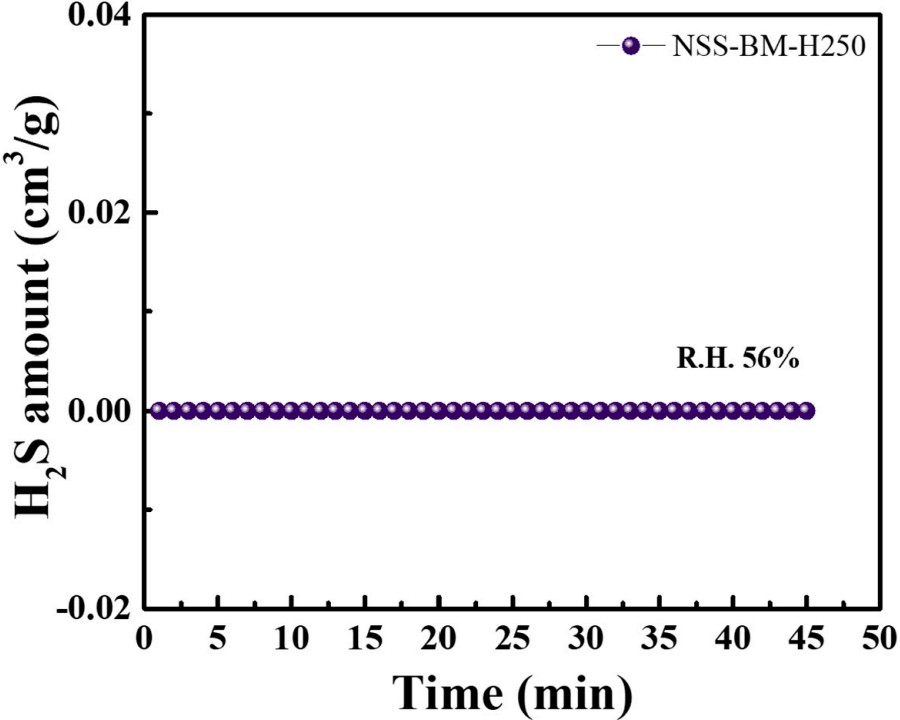

**Figure 8.** $H_2S$ concentration of $Na_3SbS_4$-BM-H250 hydrolysis in air as a function of time.

Figure 9a–d illustrate the electrochemical performance of the NFMO ($Na_{2/3}Fe_{1/2}Mn_{1/2}O_2$) cathode | $Na_3SbS_4$ -BM-H250 solid electrolyte | Na foil at RT. In Figure 9a, a comprehensive C-rate test was conducted, revealing specific capacities achieved at various current densities: 106 mAh/g at 0.01 A/g, 90 mAh/g at 0.02 A/g, 67 mAh/g at 0.05 A/g, and 98 mAh/g upon reverting to 0.01 A/g. This unveils the system's substantial charge storage potential at lower current densities, with a capacity reduction at higher densities and some recovery upon returning to the initial density. Figure 9b depicts charge and discharge cycles at current densities of 0.01 A/g, 0.02 A/g, and 0.05 A/g. The specific discharge capacities are 106 mAh/g, 90 mAh/g, and 67 mAh/g, respectively. These outcomes elucidate the

discharge behavior and its variation with different current densities. Complementing the findings in Figure 9c, Figure 9d shows the galvanostatic charge and discharge cycles at specific cycles (1 to 3, 5, 10, 20, 30). These voltage profiles visually represent the system's electrochemical behavior and confirm its stability and consistency throughout the tested cycles. Overall, the remarkable cycle life performance, sustained discharge specific capacity, and high columbic efficiency observed in Figure 9c,d highlight the potential of the NFMO and $Na_3SbS_4$ solid electrolyte system for practical energy storage applications. The system's ability to deliver reliable and efficient charge storage over an extended duration makes it a promising candidate for future advancements in energy storage technologies, aligning with the sustainability goals of modern energy solutions. This is particularly significant in the context of solid electrolytes, as evidenced by the comparison with other literature in Table 3, which highlights the performance of our work in relation to previous studies.

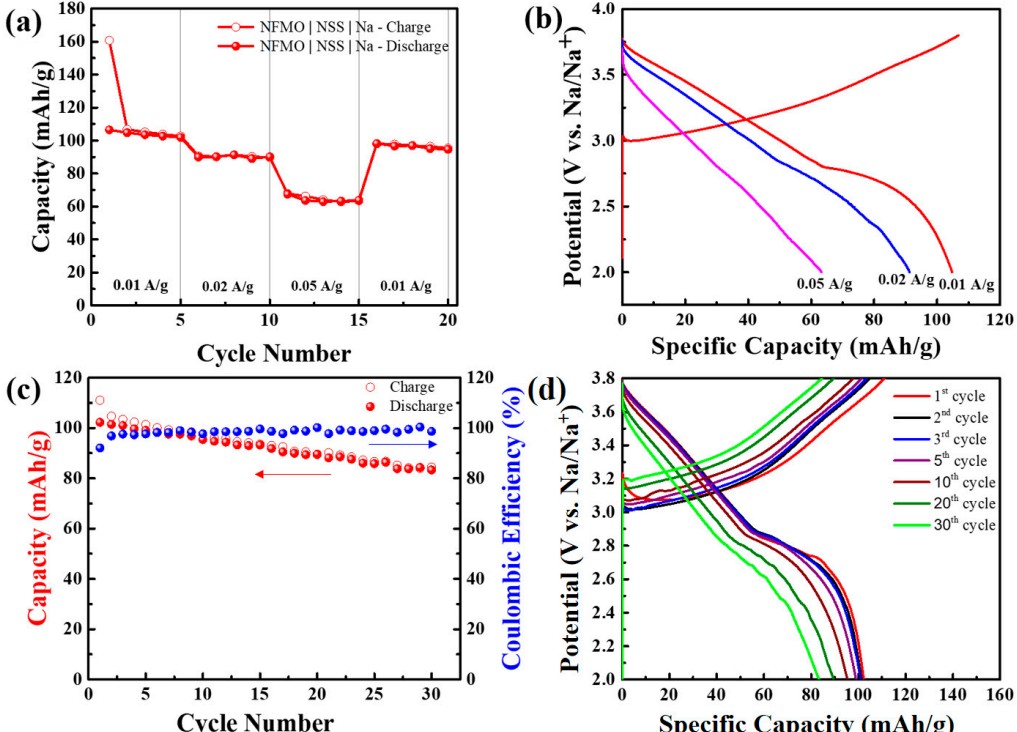

**Figure 9.** Cycling performance of $Na_{2/3}Fe_{1/2}Mn_{1/2}O_2$ | $Na_3SbS_4$-BM-H250 | Na at RT. (**a**) C-rate test, (**b**) galvanostatic charge and discharge cycle of different current densities, (**c**) cycle life test, and (**d**) galvanostatic charge and discharge cycle of current density at 0.01 A/g over 30 cycles.

**Table 3.** Comparison of $\sigma_{RT}$ and $E_a$ of $Na_3SbS_4$ from our results with those from prior reporters.

| Solid Electrolytes | Ionic Conductivity at RT (mS cm$^{-1}$) | Activation Energy (eV) | References |
|---|---|---|---|
| $Na_3SbS_4$ | 0.31 | 0.21 | This work |
| $Na_3SbS_4$ | 0.54 | 0.074 | [24] |
| $Na_3SbS_4$ | 0.31 | 0.2 | [27] |
| t-$Na_3SbS_4$ | 0.6 | 0.224 | [28] |
| $Na_3SbS_4$ | 1.06 | 0.21 | [26] |
| t-$Na_3SbS_4$ | 3 | 0.25 | [18] |
| t-$Na_3SbS_4$ | 0.8 | 0.3 | [22] |
| $Na_3SbS_4$ | 1.05 | 0.22 | [29] |

## 4. Conclusions

In this study, we successfully synthesized the tetragonal phase of the $Na_3SbS_4$ (NSS) solid electrolyte using two distinct methodologies. The second method (BM-H) is particu-

larly significant, which was executed at a temperature of 250 °C and yielded outcomes of remarkable importance. This approach has rendered a conspicuous ionic conductivity of 0.31 mS cm$^{-1}$ at room temperature while exhibiting a remarkably diminutive activation energy of 0.21 eV. Moreover, it maintains stability in the air. The $Na_3SbS_4$ synthesized through the BM-H process at 250 °C has effectively functioned as a solid electrolyte, assuming the role of an intermediary amid the $Na_{2/3}Fe_{1/2}Mn_{1/2}O_2$ (NFMO) electrode and the sodium counter electrode. This arrangement showcased a significant discharge capacity of 104 mAh/g at a current density of 0.01 A/g. Furthermore, an impressive coulombic efficiency of 99.3% was achieved over 30 cycles, thereby highlighting the efficiency of charge transfer processes. The result indicates that $Na_3SbS_4$ stands as a preeminent solid electrolyte option for sodium-ion batteries, aligning with the sustainable energy storage solutions. Moving forward, it is recommended to focus on optimizing the BM-H (second method) synthesis process for large-scale production, fine-tuning the ball milling time to control grain size, and exploring alternative characterization techniques that are more aligned with the specific properties of $Na_3SbS_4$. Additionally, modifications to the electrode design should be considered for improved interface compatibility, and extended cycling tests are crucial for assessing long-term stability. Safety assessments and cost-effective production methods should also be explored, and collaborative research with experts in relevant fields can provide invaluable insights for further development and application of $Na_3SbS_4$ solid electrolytes.

**Supplementary Materials:** The following supporting information can be downloaded at: https://www.mdpi.com/article/10.3390/su152115662/s1.

**Author Contributions:** C.B.T. and C.-H.H. wrote this paper and analyzed the data; Y.A.G. and W.-R.L. conceived and designed the experiments; C.-T.H. analyzed the data; C.-T.H. and W.-R.L. supervised and revised the manuscript. All authors have read and agreed to the published version of the manuscript.

**Funding:** This research received no external funding.

**Institutional Review Board Statement:** Not applicable.

**Informed Consent Statement:** Informed consent was obtained from all subjects involved in the study.

**Data Availability Statement:** All data are available within the article and in the supplementary materials archive.

**Acknowledgments:** The authors gratefully acknowledged to National Science of Technology Council (NSTC) project grant no. NSTC 112-2218-E-007-023, 111-2622-E-033-007, 111-2923-E-006-009, 112-2923-E-006-004 and 111-2221-E-033-004-MY3.

**Conflicts of Interest:** The authors declare no conflict of interest.

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
