# Peer review of "Synthesis and Characterization of Na3SbS4 Solid Electrolytes via Mechanochemical and Sintered Solid-State Reactions: A Comparative Study"

_sustainability, doi:10.3390/su152115662_

Round 1

Reviewer 1 Report

Comments and Suggestions for Authors

This manuscript Celastin Bebina Thairiyarayar, Chia-Hung Huang,Yasser Ashraf Gandomi, Chien-Te Hsieh, and Wei-Ren Liue entitled ‘Synthesis and characterization of Na3SbS4 solid electrolytes via mechanochemical and sintered solid-state reactions: A comparative study’ used a combination of ball milling and sintering methods to develop a non-organic solid-state electrolyte, Na3SbS4. Then it was used as a solid electrolyte with promising electrochemical performance. I recommend that this manuscript can be published in Sustainability after addressing the issues below:

1.      In the introduction part, streamlining the information in line with this research's specific aims would strengthen the introduction's clarity and relevance. While the historical background and previous research are essential, ensure that the information remains focused on the context of this study.

2.      The authors are encouraged to check the whole manuscript to ensure that any acronyms or symbols, such as "RT" for room temperature in the abstract, are defined for clarity.

3.      The author chose 20 hours of ball milling time as optimal; any longer milling times should be included in the experiment.

4.      Why is there such a significant disparity between the particle size obtained from XRD and the particle size obtained from SEM?

5.      Ensure consistency in the use of terminology, especially in the naming of the synthesized product. Using a consistent naming convention will prevent confusion and aid in clear communication. Is the NSS-BM-20h-H300 the same as NSS-BM-H300? Is the NSS-BM-20h-H250 the same as NSS-BM-H250?

6.      The samples used to measure the electrochemical performance are all sintered samples. Therefore, it is recommended that the author move the discussion regarding ball milling time to the Supplementary Information (SI) section. This adjustment would enhance the clarity of the article.

7.      What were the temperature and humidity during the measurements for assessing the stability of sulfides? The author may consider referencing the literature to ensure reliability Energy Environ. Sci., 2022,15, 991-1033.

8.      The samples sintered at 250 degrees Celsius exhibited the best performance, while samples sintered at lower temperatures need to be included for a comprehensive analysis.

9.      The Cyclic voltammogram of electrolyte are encouraged to be added in the manuscript.

10.   The author needs to refer to more recent research findings, especially those from the last five years, to ensure the incorporation of the latest developments in the field.

Comments on the Quality of English Language

Please conduct a thorough check for grammar and punctuation errors in the manuscript.

Author Response

Reviewer 1

This manuscript Celastin Bebina Thairiyarayar, Chia-Hung Huang,Yasser Ashraf Gandomi, Chien-Te Hsieh, and Wei-Ren Liue entitled ‘Synthesis and characterization of Na3SbS4 solid electrolytes via mechanochemical and sintered solid-state reactions: A comparative study’ used a combination of ball milling and sintering methods to develop a non-organic solid-state electrolyte, Na3SbS4. Then it was used as a solid electrolyte with promising electrochemical performance. I recommend that this manuscript can be published in Sustainability after addressing the issues below:

  1. In the introduction part, streamlining the information in line with this research's specific aims would strengthen the introduction's clarity and relevance. While the historical background and previous research are essential, ensure that the information remains focused on the context of this study.

Response:

Thank you for your valuable comments and constructive feedback. We greatly appreciate your thoughtful insights. We have carefully considered your suggestions and incorporated them into the revised manuscript.

  1. The authors are encouraged to check the whole manuscript to ensure that any acronyms or symbols, such as "RT" for room temperature in the abstract, are defined for clarity.

Response:

Thank you for bringing this to our attention. We sincerely apologize for the oversight. You are absolutely correct; 'RT' indeed stands for room temperature. We have taken immediate action to rectify this oversight in our revised manuscript. All acronyms and symbols, including 'RT', have been diligently defined throughout the entire manuscript to ensure clarity and comprehension for our readers.

  1. The author chose 20 hours of ball milling time as optimal; any longer milling times should be included in the experiment.

Response:

Thank you for your valuable suggestion. We have taken your advice into consideration and conducted experiments with a 25-hour ball milling time. The results of these experiments have been included in the supplementary material accompanying our revised manuscript. We have carefully analyzed the data and incorporated the findings into our discussion, providing a more comprehensive understanding of the material behavior under extended milling durations.

We sincerely appreciate your insightful input, which has significantly enriched the depth of our study.

Figure S2. (a) X-ray diffraction patterns of only ball milling (BM) with respective sample photos and (b) Nyquist impedance plots of Na3SbS4 –BM -20 h and 25 h at RT, Insect: Equivalent circuit in the system.

The following explanation has been included in the revised version of the manuscript.

Supplementary Figure 2(a) displays the X-ray diffraction (XRD) patterns and corresponding photographs of samples subjected to 20 and 25 hours of ball milling. An interesting observation is the emergence of distinct 2θ peaks around 16.6 o in BM-25 h indicating the presence of NaSbS2. This suggests a transformation or phase transition within the material due to the extended ball milling process. This alteration may be attributed to enhanced structural rearrangements and increased defects induced by prolonged milling, leading to the formation of NaSbS2. In contrast, at the 25-hour mark (Supplementary Figure 2), there is a notable surge in total resistance from 202 Ω to 403 Ω, resulting in a significant drop in ionic conductivity from 2.77 to 1.32×10-4 S cm-1. This decline can be ascribed to the prolonged milling process, which encourages particle agglomeration and carries the potential for crystal damage. Consequently, this leads to an overall increase in total resistance and a corresponding decrease in ionic conductivity.

  1. Why is there such a significant disparity between the particle size obtained from XRD and the particle size obtained from SEM?

Response:

We are sincerely grateful for your thoughtful question regarding the particle size analysis. Your keen observation has added a valuable perspective to our research. We truly appreciate the time and effort you have dedicated to reviewing our work.

The difference in particle size­s observed between XRD and SEM can be attributed to the­ inherent capabilities and me­asurement methods of the­se two techniques

In our study, we extensively analyzed the particle sizes of Na3SbS4 using both XRD and SEM techniques. The XRD analysis revealed crystallite sizes between 196 and 273 nm, which varied depending on the specific synthesis conditions and processing steps. These estimations were made based on the diffraction patterns and the application of the Scherer equation, which provided valuable insights into the internal structure of the crystallites. In contrast, scanning electron microscopy (SEM) provided high-resolution surface images that allowed for direct measurement of the particle sizes. For example, after 20 hours of ball milling, SEM revealed particle sizes ranging around 1 micrometer. After sintering at 250 °C, the particles increased in size significantly to a range of 10 to 16 micrometers. This difference between XRD and SEM measurements clearly demonstrates that SEM captures the external dimensions of the particles more accurately, providing a more precise representation of their surface size. To illustrate this disparity, let's consider the comparison of a sample that underwent 20 hours of ball milling and sintering at 250 °C. XRD analysis estimated the crystallite size to be around 273 nanometers. However, SEM analysis of the same sample revealed a particle size­ range between 10 and 16 mm, which is significantly larger. Our research has yielded practical numerical examples that clearly show the distinction between XRD and SEM when it come­s to particle size measurement. XRD tends to provide estimates of the internal crystallite­ size, while SEM measures the actual external particle size. The difference arises from the different measurement methodologies used by each technique, highlighting how XRD and SEM compleme­nt each other in material characterization.

  1. Ensure consistency in the use of terminology, especially in the naming of the synthesized product. Using a consistent naming convention will prevent confusion and aid in clear communication. Is the NSS-BM-20h-H300 the same as NSS-BM-H300? Is the NSS-BM-20h-H250 the same as NSS-BM-H250?

Response:

Thank you for bringing this matter to our attention. Your astute observation has been invaluable in clarifying the nomenclature in our study. We sincerely apologize for any confusion caused. Upon careful review, we confirm that NSS-BM-20h-H300 is indeed synonymous with NSS-BM-H300, as is NSS-BM-20h-H250 with NSS-BM-H250. To address this, we have made the necessary revisions in the experimental section of the manuscript for consistency. Your keen eye for detail has greatly contributed to the accuracy and precision of our work. We genuinely appreciate your diligence in evaluating our research.

  1. The samples used to measure the electrochemical performance are all sintered samples. Therefore, it is recommended that the author move the discussion regarding ball milling time to the Supplementary Information (SI) section. This adjustment would enhance the clarity of the article.

Response:

Thank you for your thoughtful suggestion. We appreciate your attention to detail. In response to your recommendation, we have carefully considered the placement of the discussion regarding ball milling time. After careful deliberation, we believe it is pertinent to retain this discussion in the main body of the manuscript. This allows us to maintain a clear and coherent narrative of the synthesis process and its impact on the electrochemical performance. However, to provide a more comprehensive understanding, we have included supplementary information detailing the ball milling process for interested readers. We hope this approach effectively balances clarity with the need for comprehensive information. If you have any additional feedback or specific areas you believe require further attention, please feel free to share. Your insights play a crucial role in enhancing the overall quality of our work.

Figure S1. (a) Nyquist impedance plots at RT, Insect: Equivalent circuit in the system and (b) Arrhenius plots at 25, 40, 60, 80 °C of NSS-BM10 h, 15 h, 20 h.

Table S1. Comparison of ionic conductivity and activation energy of Na3SbS4 with only BM samples from first method.

    Samples

 L

(cm)

 S

(cm2)

Rb

(Ω)

 Rgb

 (Ω)

Rb+Rgb

 (Ω)

  σ25 °C

(S cm-1 )

   Ea

  (eV)

 NSS-BM-10h

0.044

0.7854

 48

 248

 296

 1.89×10-4

  0.24

 NSS-BM-15h

0.044

0.7854

 45.9

213.9

 253.9

 2.16×10-4

  0.23

 NSS-BM-20h

0.044

0.7854

  31

171.4

 202.4

 2.77×10-4

  0.22

Explanation from revised manuscript,

Supplementary Figure 1(a) offers a clear depiction of the ionic conductivity plot, showcasing the effects of varying ball milling durations (10h, 15h) on NSS samples. As the duration of ball milling increases, there is a marked reduction in total resistance. This reduction reaches its zenith at the 20-hour mark, resulting in a substantial boost in ionic conductivity. These enhancements can be attributed to the refinement in crystal structure and the diminishment of particle size. Together, these elements synergize to amplify ion mobility and effectively suppress resistance. Supplementary Figure 1(b) complements this observation by presenting the activation energy corresponding to ball milling durations of 10h, 15h, and 20h. It's worth noting that an extension in ball milling time leads to a reduction in activation energy, signifying an improvement in ionic conductivity. For a detailed breakdown of the ionic conductivity and activation energy values, please refer to Supplementary Table 1.

  1. What were the temperature and humidity during the measurements for assessing the stability of sulfides? The author may consider referencing the literature to ensure reliability (Energy Environ. Sci., 2022,15, 991-1033).

Response:

Thank you for your inquiry regarding the measurement conditions for assessing the stability of sulfides. We greatly value your attention to detail. The measurements were conducted under controlled conditions with a temperature of [ room temperature (RT)] and a humidity level of [56%]. To further fortify the reliability of our results, we have heeded the recommendations provided in the referenced literature (Energy Environ. Sci., 2022, 15, 991-1033). This step ensured that our experimental setup adheres to widely recognized standards within the field. Your suggestion has significantly contributed to the robustness of our study, and we are appreciative of your expertise in this matter. Additionally, we have incorporated this information into our revised manuscript.

Here is the reference,

  1. Nikodimos, Y.; Huang, C. J.; Taklu, B. W.; Su, W. N.; Hwang, B. J. Chemical stability of sulfide solid-state electrolytes: stability toward humid air and compatibility with solvents and binders. Energy & Environmental Science, 2022, 15(3), 991-1033.

  1. The samples sintered at 250 degrees Celsius exhibited the best performance, while samples sintered at lower temperatures need to be included for a comprehensive analysis.

Response:

Thank you for your valuable feedback. We appreciate your suggestion to include samples sintered at lower temperatures for a more comprehensive analysis. We would like to inform you that we have already incorporated the results and findings from samples sintered at 150 and 200 °C s in the supplementary section. However, it's important to note that we observed lower ionic conductivity and the presence of impurities, particularly at 150 °C. Based on these findings, we concluded that the sintering temperature of 250 °C yielded the best performance, which is why we chose this temperature for our primary analysis. If you have any further suggestions or specific aspects you would like us to explore, please feel free to let us know. Your input is greatly valued in enhancing the quality of our work.

 Figure S3. (a) X-ray diffraction patterns of BM and then sintering Na3SbS4 (BM-H) with respective sample photos and (b) Nyquist impedance plots at RT, Insect: Equivalent circuit in the system of NSS-BM20h-H250, H200, H150 °C.

Table S2. Comparison of ionic conductivity of Na3SbS4 with only BM from first method.

    Samples

 L

(cm)

 S

(cm2)

Rb

(Ω)

 Rgb

 (Ω)

Rb+Rgb

 (Ω)

  σ25 °C

(S cm-1 )

NSS-BM20h-H150

0.044

0.7854

49.04

205.6

254.6

2.20×10-4

NSS-BM20h-H200

0.043

0.7854

43.81

148.8

192.6

2.26×10-4

NSS-BM20h-H250

0.043

0.7854

11.43

164.7

176.13

3.11×10-4

Here is the reason we attached in our revised version, 

Supplementary Figure 3(a) showcases the X-ray diffraction (XRD) patterns along with accompanying photographs of NSS-BM20h-H150, 200, and 250 samples. Notably, the XRD pattern of NSS-BM20h-H150 exhibits several minor impurity peaks. This occurrence can be attributed to the lower sintering temperature (150 °C), which may not provide sufficient energy to promote complete phase transformation and crystallization. Consequently, residual impurities persist in the sample, leading to the observed minor peaks in the XRD pattern. Supplementary Figure 3 (b) imparts valuable insights into the impact of sintering temperature on ionic conductivity, with a specific focus on NSS BM-20h samples sintered at 150 °C, 200 °C, and 250 °C. A conspicuous trend emerges: as sintering temperature diminishes, so does ionic conductivity. This phenomenon finds its roots in the diminished thermal energy at lower temperatures, which effectively curtails ion mobility, consequently resulting in a discernible reduction in ionic conductivity. Detailed ionic conductivity values are provided in Supplementary Table 2.

  1. The Cyclic voltammogram of electrolyte are encouraged to be added in the manuscript.

Response:

We sincerely appreciate your valuable suggestion regarding the inclusion of cyclic voltammogram data in our manuscript. We have given careful consideration to your recommendation and would like to provide additional context on why we did not incorporate this specific test.

It is worth noting that our study aligns with several other published works in this field (references 22, 24, 28) which have successfully provided meaningful insights without the inclusion of cyclic voltammetry (CV) tests. Additionally, we believe that the extensive array of experiments and analyses detailed in the manuscript provides a robust characterization of the materials and system under investigation. The results presented offer substantial insights into the electrochemical performance of the Na2/3Fe1/2Mn1/2O2 (NFMO) cathode paired with the Na3SbS4 solid electrolyte, demonstrating its potential for practical energy storage applications.

Moreover, we found substantial support for our findings in the existing literature. For instance, reference (26) discusses the electrochemical stability of a similar material and highlights that the cyclic performance of the full cell was not significantly influenced by the electrochemical behavior within the 1.2–2.4 V range. Similarly, reference (18) evaluates the electrochemical stability of Na3SbS4 with metallic sodium using CV measurements and provides valuable insights into the system's behavior.

We hope this information provides further clarity on our approach. If you have any additional questions, suggestions, or alternative tests you believe would be beneficial, we remain open to considering them. Thank you once again for your insightful feedback.

 Here are the references about cyclic voltammograms of electrolyte,

  1. Wu, E. A.; Kompella, C. S.; Zhu, Z.; Lee, J. Z.; Lee, S. C.; Chu, I. H.; & Meng, Y. S. New insights into the interphase between the Na metal anode and solid-state sulfide electrolytes: a joint experimental and computational study. ACS Applied Materials & Interfaces, 2018, 10(12), 10076-10086.

  1. Li, Y.; Arnold, W.; Halacoglu, S., Jasinski, J. B.; Druffel, T.; & Wang, H. Phase‐Transition Interlayer Enables High‐Performance Solid‐State Sodium Batteries with Sulfide Solid Electrolyte. Advanced Functional Materials, 2021, 31(28), 2101636.

  1. Zhang, L.; Zhang, D.; Yang, K.; Yan, X.; Wang, L.; Mi, J.; Li, Y. Vacancy‐contained tetragonal Na3SbS4 superionic conduc-tor. Advanced Science, 2016, 3(10), 1600089.

  1. Rush Jr, L. E.; Hood, Z. D.; & Holzwarth, N. A. W. Unraveling the electrolyte properties of Na3SbS4 through computation and experiment. Physical Review Materials, 2017, 1(7), 075405.

  1. Liang, B.; Yu, L.; Wang, G.; Lin, C.; Gao, C.; Shen, X.; & Jiao, Q. Physical and electrochemical behavior of affordable Na3SbS4 solid electrolyte at different heat treatment temperatures. Ceramics International, 2022, 48(20), 30144-30150.

  1. Zhang, Q; Zhang, C.; Hood, Z. D.; Chi, M.; Liang, C.; Jalarvo, N. H.; & Wang, H. Abnormally low activation energy in cubic Na3SbS4 superionic conductors. Chemistry of Materials, 2020, 32(6), 2264-2271.

  1. Wang, H.; Chen, Y.; Hood, Z. D.; Keum, J. K.; Pandian, A. S.; Chi, M.; Sunkara, M. K. Revealing the structural stability and Na-ion mobility of 3D superionic conductor Na3SbS4 at extremely low temperatures. ACS Applied Energy Materials, 2018, 1(12), 7028-7034.

  1. The author needs to refer to more recent research findings, especially those from the last five years, to ensure the incorporation of the latest developments in the field.

Response:

Thank you for your feedback regarding the literature review. We acknowledge your concern and appreciate your perspective. In our manuscript, we aimed to strike a balance between providing a comprehensive overview of the relevant literature and focusing on the specific contributions of our work. We conducted a thorough review and ensured that our study is situated within the context of existing research, as evidenced by the comparative analysis presented in Table 3. While we understand your suggestion, we believe that the current approach effectively contextualizes our findings and highlights their significance. Additionally, we provided synthesis methods and results findings from other relevant literature in the introduction section to establish the broader research context. This approach was taken to ensure that our work is well-grounded and contributes meaningfully to the field. If you have any further suggestions or specific references you believe should be included, we would be open to considering them. Thank you for your continued engagement with our work.

Reviewer 2 Report

Comments and Suggestions for Authors

The paper “Synthesis and characterization of Na3SbS4 solid electrolytes via mechanochemical and sintered solid state reactions: A comparative study” is devoted to obtaining Na3SbS4 by two methods: ball grinding and a combination of ball milling and sintering, as well as a study of the charging cyclic durability of a system based on Na2/3Fe1/2Mn1/2O2|t-NSS|Na. XRD, FE-SEM, EDS, XPS, EIS techniques were used for the samples characterization.  The work is of scientific interest for specialists in the field synthesis of solid electrolytes. Nevertheless, there are several points before the paper can be published. I hope that authors after minor revisions can improve the paper and can publish it in Sustainability.

  1. Indentations throughout the text should be the same.
  2. Why exactly this number of revolutions per minute was used on the ball mill during the experiment (510rpm)?
  3. What causes a decrease in crystallites during sintering at 350 °C?
  4. Table captions must have the same style.
  5. Place a space between the temperature value and the degrees Celsius designation.
  6. More practical recommendations should be added in the manuscript.
  7. Explain your conclusion in more detail.
Comments on the Quality of English Language

Minor editing of English language required

Author Response

Reviewer 2

The paper “Synthesis and characterization of Na3SbS4 solid electrolytes via mechanochemical and sintered solid state reactions: A comparative study” is devoted to obtaining Na3SbS4 by two methods: ball grinding and a combination of ball milling and sintering, as well as a study of the charging cyclic durability of a system based on Na2/3Fe1/2Mn1/2O2|t-NSS|Na. XRD, FE-SEM, EDS, XPS, EIS techniques were used for the samples characterization.  The work is of scientific interest for specialists in the field synthesis of solid electrolytes. Nevertheless, there are several points before the paper can be published. I hope that authors after minor revisions can improve the paper and can publish it in Sustainability.

  1. Indentations throughout the text should be the same.

Response:

We sincerely appreciate your observant attention to detail regarding the consistency of indentations in the text. After a thorough review, we have taken steps to ensure uniformity in this aspect throughout the manuscript. These corrections have been diligently incorporated into our revised version. Your valuable input has been instrumental in enhancing the quality of our work, and we are grateful for your thorough evaluation.

  1. Why exactly this number of revolutions per minute was used on the ball mill during the experiment (510rpm)?

Response:

Thank you for your inquiry regarding the choice of ball mill speed at 510 revolutions per minute (rpm) during our experiment.

In selecting the rotational speed of the ball mill at 510 revolutions per minute (rpm), we conducted a preliminary set of experiments to determine the optimal milling conditions for synthesizing Na3SbS4. This speed was chosen based on a combination of factors, including the desired particle size distribution, material properties, and previous literature findings [23]. Additionally, we took into consideration the balance between achieving sufficient grinding efficiency and avoiding excessive mechanical stress on the material, which could lead to unwanted phase transformations or structural defects. Through systematic experimentation, we found that 510rpm provided the optimal balance between these factors, resulting in the desired particle size and crystalline structure observed in our XRD and SEM analyses.

Here is the reference,

  1. Zhang, S.; Zhao, Y.; Zhao, F.; Zhang, L.; Wang, C.; Li, X., & Sun, X. Gradiently sodiated alucone as an interfacial stabilizing strategy for solid‐state Na metal batteries. Advanced Functional Materials, 2020, 30(22), 2001118.

  1. What causes a decrease in crystallites during sintering at 350 °C?

Response:

Thank you for your insightful question regarding the decrease in crystallite size observed during sintering at 350 °C. This phenomenon can be attributed to the elevated thermal energy and extended exposure to high temperatures inherent in the sintering process. At 350 °C, the material experiences heightened atomic mobility, facilitating the reorganization and growth of smaller crystallites into larger, more ordered domains. This process is a common occurrence in materials science, where high-temperature treatments lead to grain growth and structural rearrangements.

In our specific study, we observed a decrease in crystallite size from 208 nm in NSS-BM-H350 to 273 nm in NSS-BM-H250 after subjecting the material to 20 hours of ball milling followed by sintering. This reduction aligns with the expected behavior under high-temperature conditions. It is important to note that the material's composition, impurities, and initial crystallite distribution also play a role in influencing sintering behavior. Overall, this finding underscores the significance of temperature control in manipulating the microstructure of Na3SbS4. It provides valuable insights for tailoring the material's properties for a range of applications. We deeply appreciate your astute observation, which has enriched the understanding of our research.

  1. Table captions must have the same style.

Response:

Thank you for bringing this to our attention. We sincerely appreciate your meticulous review. We apologize for the inconsistency in table captions. Rest assured, we have rectified this oversight and ensured uniform styling for all table captions. The revised manuscript now reflects this correction. Your keen eye for detail has been instrumental in enhancing the overall quality of our work.

  1. Place a space between the temperature value and the degrees Celsius designation.

Response:

Thank you for your valuable feedback. We genuinely appreciate your thorough examination of our manuscript. We apologize for the oversight in spacing between temperature values and the °C designation. This has now been rectified, and a space has been appropriately included. The revised manuscript reflects this correction. Your meticulous review has significantly contributed to the refinement of our work. We are grateful for your attention to detail.

  1. More practical recommendations should be added in the manuscript.

Response:

Thank you for your valuable feedback and suggestions. We appreciate your thorough review of our manuscript. Regarding the request for additional practical recommendations, we would like to highlight that the conclusion section of our manuscript already encompasses the key findings and implications of our study. We believe that the recommendations provided therein offer a comprehensive perspective on the potential applications and future directions of our research. We have taken into consideration your suggestion and carefully assessed the possibility of further recommendations. However, we believe that the existing conclusion adequately addresses the outcomes of our investigation. Once again, we sincerely appreciate your time and effort in reviewing our work.

  1. Explain your conclusion in more detail.

Response:

Thank you for your insightful comment. We appreciate your suggestion to elaborate on the conclusion. We have duly considered this and, in our revised manuscript, we have provided a more comprehensive explanation of the study's findings. Your valuable input has greatly contributed to the enhancement of our work. We are grateful for your time and dedication in reviewing our research.

Reviewer 3 Report

Comments and Suggestions for Authors

This is a very interesting research. The work is well-structured, has coherence, scientific soundness, and is clearly written. The literature revision covers an extended period of time and is correctly used in the analysis of the results.

  •  

  • The paper can be published practically as it is.

    Only there are two observations that I think can be corrected in the edition process:

    - The text in the images of Figure 2 is unclear, especially the identification letters.

    - In Table 1, the first column heading needs to be corrected.

  •  

Author Response

Reviewer 3

This is a very interesting research. The work is well-structured, has coherence, scientific soundness, and is clearly written. The literature revision covers an extended period of time and is correctly used in the analysis of the results.

  • The paper can be published practically as it is.

Only there are two observations that I think can be corrected in the edition process:

- The text in the images of Figure 2 is unclear, especially the identification letters.

Response:

Thank you for bringing this to our attention. We apologize for any inconvenience caused by the unclear text in Figure 2. We have already addressed this issue in our revised manuscript. To provide further clarity, we have also attached an updated version of Figure 2 for your reference. Thank you for your thorough review.

Figure 2. SEM images of (a) NSS precursor (NSS-BM-0h), (b) NSS-BM-10 h, c) NSS-BM-15 h and d) NSS-BM-20 h.

- In Table 1, the first column heading needs to be corrected.

Response:

Thank you for bringing this to our attention. We apologize for the oversight. We have already rectified the first column heading in our revised manuscript. Here are the corrected table headings for your reference:

Table 1. Crystal structure information of Na3SbS4 –BM20h-H250.

Table 2. Comparison of ionic conductivity and activation energy of Na3SbS4 with BM and BM and then Sintering at different temperature.

Table 3. Comparison of σRT and Ea of Na3SbS4 from our results with those from prior reporters.

Round 2

Reviewer 1 Report

Comments and Suggestions for Authors

The authors' responses to the reviewers' comments were thorough and satisfactory. As such, this manuscript is acceptable for publication without further changes required.